# Tubeimoside-1 Enhances TRAIL-Induced Apoptotic Cell Death through STAMBPL1-Mediated c-FLIP Downregulation

**DOI:** 10.3390/ijms241411840

**Published:** 2023-07-24

**Authors:** So Rae Song, Seung Un Seo, Seon Min Woo, Ji Yun Yoon, Simmyung Yook, Taeg Kyu Kwon

**Affiliations:** 1Department of Immunology, School of Medicine, Keimyung University, Daegu 42601, Republic of Korea; ssr02067@naver.com (S.R.S.); sbr2010@hanmail.net (S.U.S.); woosm724@gmail.com (S.M.W.); libra3009@naver.com (J.Y.Y.); 2College of Pharmacy, Keimyung University, Daegu 42601, Republic of Korea; ysimmyung@kmu.ac.kr; 3Center for Forensic Pharmaceutical Science, Keimyung University, Daegu 42601, Republic of Korea

**Keywords:** TBMS-1, c-FLIP, STAMBPL1, TRAIL, deubiquitinase

## Abstract

Tubeimoside-1 (TBMS-1), a traditional Chinese medicinal herb, is commonly used as an anti-cancer agent. In this study, we aimed to investigate its effect on the sensitization of cancer cells to tumor necrosis factor-related apoptosis-inducing ligand (TRAIL). Our results revealed that even though monotherapy using TBMS-1 or TRAIL at sublethal concentrations did not affect cancer cell death, combination therapy using TBMS-1 and TRAIL increased apoptotic cell death. Mechanistically, TBMS-1 destabilized c-FLIP expression by downregulating STAMBPL1, a deubiquitinase (DUB). Specifically, when STAMBPL1 and c-FLIP bound together, STAMBPL1 deubiquitylated c-FLIP. Moreover, STAMBPL1 knockdown markedly increased sensitivity to TRAIL by destabilizing c-FLIP. These findings were further confirmed in vivo using a xenograft model based on the observation that combined treatment with TBMS-1 and TRAIL decreased tumor volume and downregulated STAMBPL1 and c-FLIP expression levels. Overall, our study revealed that STAMBPL1 is essential for c-FLIP stabilization, and that STAMBPL1 depletion enhances TRAIL-mediated apoptosis via c-FLIP downregulation.

## 1. Introduction

Tubeimoside-1 (TBMS-1) is a triterpenoid saponin present in the Chinese medicinal herb *Bolbostemma paniculatum* (Maxim) Franquet (Cucurbitaceae) [1]. It exerts several therapeutic effects, including anti-inflammatory, anti-angiogenesis, and anti-cancer [2,3,4].

Jiang et al. reported that TBMS-1 enhances apoptosis in breast cancer via dephosphorylated Akt-mediated downregulation of anti-apoptotic Bcl-2 proteins, such as Mcl-1, Bcl-2, and Bcl-xL [5]. Other studies have also shown that it induces cancer cell death through the impairment of autophagic flux in breast, cervical, and colorectal cancer cells [6,7]. Further, TBMS-1 can initiate autophagy via AMPK-dependent autophagosome formation. Notably, it accumulates impaired autophagolysosomes by blocking autophagic flux via the inactivation of lysosomal enzymes, and this enhances the efficacy of chemotherapeutic drugs, including cisplatin, doxorubicin, and 5-fluorouracil in cervical and colorectal cancer [6,7]. Recently, Xiao et al. reported that TBMS-1 exerts anti-cancer effects by mediating the crosstalk between mitochondria and lysosomes [8]. They further suggested that TBMS-1 inhibits autophagic flux by inhibiting lysosomal acidification and by increasing reactive oxygen species (ROS) level via mitochondrial fragmentation in lung cancer. Thus, generated ROS increases lysosomal membrane permeabilization. This results in the release of cathepsin B to the cytoplasm, which eventually causes cytochrome *c*-mediated caspase-dependent apoptosis via a positive feedback loop [8].

Tumor necrosis factor (TNF)-related apoptosis-inducing ligand (TRAIL) activates the caspase-dependent apoptotic cascade by binding to death receptors 4 and 5 (DR4 and DR5, respectively) [9]. There are two types of TRAIL signaling pathways. In the first (type I), activated caspase-8 directly increases cell death through effector caspases, such as caspase-3 and -7. In the second (type II), BID cleavage by caspase-8 induces mitochondrial membrane permeabilization and the release of cytochrome *c*; hence, there is increased apoptosis [10].

Owing to the selective ability of TRAIL to increase cell death in cancer, it is frequently used as an anti-cancer drug [11]. Specifically, two types of pharmacological agents, such as recombinant TRAIL and DR agonists, are used to enhance the efficacy of TRAIL-based therapy. Clinical trials based on monotherapy or combination therapy using these agents have confirmed their safety and efficacy [12,13]. However, some cancer cells are resistant to TRAIL owing to the downregulation of the expression levels of DRs or the upregulation of anti-apoptotic proteins, such as Bcl-2 and IAP family proteins [14]. Therefore, several researchers have focused on the development of new TRAIL sensitizers and on overcoming TRAIL resistance [15].

Cellular FLICE-like inhibitory protein (c-FLIP) negatively regulates DR-mediated apoptosis by interrupting DISC formation. Considering that a homolog of c-FLIP is similar to caspase-8, it can be recruited to DISC by competing with pro-caspase-8, thereby inhibiting apoptosis [16]. Therefore, c-FLIP overexpression accelerates cancer cell growth and tumor progression [17,18]. c-FLIP protein is regulated through the ubiquitin–proteasome system (UPS) at the post-translational level. STAM-binding-protein-like 1 (STAMBPL1), a deubiquitinase (DUB), is one of the regulators capable of involving c-FLIP degradation [19]. Additionally, the suppression of STAMBPL1 inhibits tumor growth and the proliferation in colorectal and gastric cancer cells [20,21]. Therefore, STAMBPL1 may be an oncogene in cancer. 

In this study, we aimed to investigate the effect of TBMS-1 on TRAIL sensitization in cancer cells as well as the underlying molecular mechanisms. Our results indicated that TBMS-1 degraded c-FLIP protein in a STAMBPL1-dependent manner, resulting in an increase in TRAIL-mediated apoptotic cancer cell death.

## 2. Results

### 2.1. TBMS-1 Increases TRAIL Sensitivity in Various Cancer Cells

Reportedly, TBMS-1 at high concentrations (e.g., 20 μM) exerts anti-cancer effects against various cancer cell lines [22,23]. In this study, we investigated whether it enhances TRAIL-induced apoptosis in human carcinoma cells. Our results revealed that TBMS-1 (5 μM) or TRAIL (50 ng/mL) monotherapy had no effect on apoptosis in human renal carcinoma Caki, colon carcinoma HCT116, lung carcinoma A549, and cervical carcinoma HeLa cells (Figure 1A). Interestingly, combination therapy using TBMS-1 and TRAIL remarkably increased the sub-G1 population and PARP cleavage in cancer cells. However, this combination therapy did not affect the morphology of apoptotic bodies or enhance sub-G1 accumulation in normal human umbilical vein EA.hy926 cells (Figure 1B). Additionally, the cancer cells showed nuclear condensation and DNA fragmentation following this combination treatment (Figure 1C,D). To verify whether the caspase is involved in cancer cell death by TBMS-1 and TRAIL treatment, we pretreated the cancer cells with pan-caspase inhibitor z-VAD-fmk (z-VAD). We observed that apoptosis induced by the combined treatment is markedly inhibited by z-VAD (Figure 1E), suggesting that TBMS-1 sensitizes TRAIL-mediated apoptosis in cancer cells.

### 2.2. TBMS-1-Induced c-FLIP Downregulation Is Involved in TRAIL Sensitization

The examination of the expression levels of apoptosis-related proteins in TBMS-1-treated cells showed that TBMS-1 significantly downregulated survivin and c-FLIP expression levels (Figure 2A). Further, to explore the role of survivin and c-FLIP in apoptosis with regard to the combination therapy, we established cells with survivin and c-FLIP overexpression. We observed that only ectopic c-FLIP, but not survivin-overexpressed cells, reduced the sub-G1 population and PARP cleavage via the combinations treatment with TBMS-1 and TRAIL (Figure 2B,C). These observations confirmed that c-FLIP downregulation plays an essential role in TBMS-1-induced TRAIL sensitization.

### 2.3. TBMS-1-Induced c-FLIP Downregulation Is Regulated by The Ubiquitin–Proteasome Pathway

We investigated alterations in mRNA levels to explore c-FLIP regulation at the transcriptional level. We observed that TBMS-1 did not alter the c-FLIP mRNA level (Figure 3A). Next, the examination of c-FLIP stabilization using cycloheximide (CHX) to confirm the relevance of post-translational regulation showed that TBMS-1 as well as CHX alone decreased c-FLIP protein levels after 3 h (Figure 3B,C). Moreover, combined treatment using CHX and TBMS-1 further accelerated the downregulation of c-FLIP protein after 1 h (Figure 3C). Our results also indicated that the proteasome inhibitor MG132 inhibited TBMS-1-mediated c-FLIP downregulation (Figure 3D). Reportedly, c-FLIP protein stability is regulated by the UPS, which induces the ubiquitination of target proteins via the attachment of ubiquitin chains, thereby degrading the proteins [24]. Therefore, as expected, TBMS-1 enhanced c-FLIP ubiquitination (Figure 3E). Collectively, these findings indicate that TBMS-1-induced c-FLIP downregulation is modulated via proteasome activation.

### 2.4. STAMBPL1 Is Critical to TBMS-1-Mediated c-FLIP Downregulation 

As diverse E3 ligases and DUBs are involved in c-FLIP protein stabilization [25], we investigated whether E3 ligases and DUBs are capable of regulating c-FLIP in TBMS-1-treated cells. In the case of E3 ligases, we observed that TBMS-1 treatment slightly decreased Cbl (Casitas B-lineage lymphoma), but had no effect on Itch (Figure 4A). However, TBMS-1 significantly downregulated the expression of STAMBPL1, a DUB of c-FLIP, but had no effect on other DUBs (USP2, USP8, and USP9x) (Figure 4A). Additionally, TBMS-1 treatment downregulated c-FLIP and STAMBPL1 expression in HeLa and A549 cells; however, such effects were absent in normal EA.hy926 cells (Figure 4B). As DUBs remove ubiquitin chains by binding to the target protein and inhibiting ubiquitination, we performed an immunoprecipitation (IP) assay using the STAMBPL1 antibody and observed that endogenous STAMBPL1 interacts with c-FLIP (Figure 4C).

Further, the exploration of the regulatory mechanisms associated with TBMS-1-mediated STAMBPL1 downregulation did not show any alteration of STAMBPL1 mRNA levels by TBMS-1 (Figure 4D). Moreover, TBMS-1- or CHX-only treatments downregulated STAMBPL1 protein expression levels from 9 h (Figure 4E,F). Particularly, pretreatment with CHX rapidly enhanced the degradation of STAMBPL1 by TBMS-1 (Figure 4F). Therefore, TBMS-1 downregulated STAMBPL1 at the post-translational regulation level. Further, we demonstrated the involvement of STAMBPL1 in combined treatment-induced apoptosis using an overexpression or depletion system. STAMBPL1 overexpression markedly prevented apoptosis induced by TBMS-1 and TRAIL combination therapy as well as c-FLIP degradation (Figure 4G). Conversely, STAMBPL1 knockdown enhanced sensitivity to TRAIL (Figure 4H). Collectively, our findings suggested that STAMBPL1 contributes to c-FLIP degradation and TRAIL sensitization by TBMS-1.

### 2.5. Combined Treatment with TBMS-1 and TRAIL Suppresses Tumor Growth In Vivo

We investigated the synergistic effects of the TBMS-1 and TRAIL combination therapy using a mouse xenograft model. The combined treatment significantly suppressed tumor growth and enhanced the TUNEL-positive signal (Figure 5A,B). Further, the examination of the levels of the proteins regulated by TBMS-1 using samples collected from mice showed downregulation of STAMBPL1 and c-FLIP expression in the TBMS-1 alone and combined treatment groups (Figure 5C). Moreover, using the TCGA dataset, we observed a shorter overall survival in patients with renal cell carcinoma (RCC) showing high c-FLIP or STAMBPL1 expression levels. Furthermore, we observed a positive correlation between c-FLIP and STAMBPL1 (Figure 5E).

## 3. Discussion

In this study, we observed that TBMS-1 enhanced TRAIL-induced apoptosis in cancer cells, and two major mechanisms were found to be involved in this process: (1) the downregulation of STAMBPL1 and (2) the proteasome-dependent degradation of c-FLIP. TBMS-1 degraded c-FLIP protein expression by decreasing STAMBPL1 protein levels, and in turn this sensitized TRAIL-mediated apoptosis (Figure 6). 

Based on pharmacokinetics, the optimum concentration and processing times of a drug determine its efficacy and toxicity, which is important in drug development [26,27]. After oral administration, TBMS-1 is degraded by the gastrointestinal tract. Thereafter, it is distributed to diverse tissues, resulting in poor absorption and limited bioavailability [28]. Alternatively, TBMS-1 has a long life-span in the body of rats owing to its low clearance by the liver after intravenous administration [29]. Therefore, its administration via intravenous injection enhances its efficacy. However, the toxicity of TBMS-1 in mice, rats, and dogs has been reported in previous studies. In mice, the LD50 values of orally and intramuscularly administered TBMS-1 were determined to be 315 and 40 mg/kg, respectively [1]. In this study, we injected mice with 10 mg/kg of TBMS-1 to verify the sensitization effect of combination therapy with TRAIL (Figure 5A–C). Therefore, TBMS-1 may be an excellent target for drug development for cancer treatment using combination therapy in the future.

Many E3 ligases and DUBs can alter the stability of c-FLIP proteins. Specifically, E3 ligase Cbl destabilizes c-FLIP and increases sensitivity to TRAIL [30]. Moreover, Itch ubiquitinates c-FLIP and causes its proteasome-dependent degradation [31]. In this study, our findings showed that TBMS-1 slightly inhibits Cbl expression (Figure 4A). Further, considering that the activation of Cbl affects c-FLIP ubiquitination and degradation based on our results, Cbl may not be associated with c-FLIP destabilization by TBMS-1. Therefore, we focused on the expression levels of DUB proteins in TBMS-1-treated cells. The c-FLIP protein is directly or indirectly stabilized by various DUBs, such as USP2, USP8, USP9X, and STAMBPL1 [19,32,33,34]. Interestingly, the results of this study showed that TBMS-1 only downregulates STAMBPL1 expression (Figure 4A,B). Additionally, we observed that STAMBPL1 endogenously interacts with c-FLIP (Figure 4C). We previously reported the involvement of STAMBPL1-mediated c-FLIP degradation in TRAIL sensitization and showed that STABMPL1 overexpression reduces c-FLIP ubiquitination [19]. In this study, we investigated the regulatory mechanism of STAMBPL1 downregulation by TBMS-1 and observed that TBMS-1 can destabilize STAMBPL1 at the post-translational level (Figure 4D−F). Therefore, some E3 ligase or DUB may be involved in TBMS-1-mediated STAMBPL1 degradation. Nevertheless, further studies are required to verify the upstream molecules of STAMBPL1. 

Chen et al. reported that the silencing of STAMBPL1 using shRNA triggers caspase-dependent death via XIAP degradation in prostate cancer cells [35]. However, as we did not observe TBMS-1-induced XIAP downregulation in this study, we ruled out the involvement of XIAP in TBMS-1-mediated TRAIL sensitization (Figure 2A). Moreover, Chen et al. indicated that STAMBPL1 depletion increases cell death 72 h after shRNA treatment; however, our data did not show STAMBPL1 siRNA-induced apoptosis after treatment for 24 h (Figure 4G). However, considering the observation that STAMBPL1 knockdown reduced c-FLIP expression, we proposed c-FLIP as a new substrate of STAMBPL1. Furthermore, STAMBPL1 depletion increased sensitivity to TRAIL owing to c-FLIP degradation (Figure 4F). 

Taken together, our findings indicated that TBMS-1 downregulates c-FLIP by downregulating STAMBPL1 deubiquitinating enzymes in various cancer cells. Therefore, TBMS-1 + TRAIL combination therapy may be considered a potential therapeutic strategy in cancer management.

## 4. Materials and Methods

### 4.1. Cell Cultures and Materials 

All the cancer cell lines used in this study (human renal carcinoma Caki, lung cancer A549, cervical cancer HeLa, and colon cancer HCT116 cells) were procured from the American Type Culture Collection (Manassas, VA, USA); human umbilical vein EA.hy926 cells were gifted by T.J. Lee (Yeungnam University, Daegu, Republic of Korea). The cells were cultured in the appropriate medium containing 10% fetal bovine serum (Welgene, Gyeongsan, Republic of Korea), 1% penicillin/streptomycin (Thermo Scientific, Waltham, MA, USA), and 100 g/mL gentamicin (Thermo Scientific) at 37 °C in a humidified atmosphere containing 5% CO_2_. TBMS-1 cells were purchased from Selleck Chem (Houston, TX, USA). Human recombinant TRAIL and z-VAD-fmk were provided by R&D Systems (Minneapolis, MN, USA). MG132 was purchased from Calbiochem (SanDiego, CA, USA). Anti-PARP, anti-cleaved caspase-3, anti-Bcl-xL, anti-Bax, anti-cIAP1, anti-DR5, and anti-USP2 were obtained from Cell Signaling Technology (Beverly, MA, USA). Anti-Bim and anti-XIAP antibodies were purchased from BD Biosciences (San Jose, CA, USA). Anti-Mcl-1, anti-Bcl-2, anti-cIAP2, anti-DR4, anti-Cbl, anti-Ub, anti-Itch, and anti-STAMBPL1 antibodies were purchased from Santa Cruz Biotechnology (Dallas, TX, USA). Anti-survivin antibody was purchased from R&D Systems. Anti-USP9X antibody was purchased from Novus Biologicals (Centennial, CO, USA). Anti-caspase-3 and anti-c-FLIP antibodies were purchased from Enzo Life Sciences (San Diego, CA, USA). Anti-USP8 antibody was purchased from Thermo Scientific. The anti-actin antibody and cycloheximide were purchased from Sigma Aldrich (St. Louis, MO, USA).

### 4.2. Investigation of Apoptosis

To investigate apoptosis, we used diverse experimental procedures. The cells were incubated with RNase for 30 min at 37 °C and added to 50 μg/mL propidium iodide (Sigma Aldrich), after fixation with 95% ethanol for 1 h at 4 °C. The sub-G1 population was analyzed using a BD Accuri™ C6 flow cytometer (BD Biosciences, San Jose, CA, USA) [36]. Nuclei condensation was observed via fluorescence microscopy (Carl Zeiss, Jena, Germany) after staining with 300 nM 4′,6′-diamidino-2-phenylindole solution (Roche, Mannheim, Germany) [37]. DNA fragmentation was then analyzed using a death detection ELISA Plus kit (Boehringer Mannheim, Indianapolis, IN, USA).

### 4.3. Western Blotting

The supernatant was collected from cells via lysing with radioimmunoprecipitation assay (RIPA) lysis buffer (20 mM HEPES and 0.5% Triton X-100, pH 7.6). Proteins were separated via SDS-PAGE and transferred to nitrocellulose membranes (GE Healthcare Life Sciences, Pittsburgh, PO, USA). Thereafter, their expression levels were detected using Immobilon Western Chemiluminescent HRP Substrate (EMD Millipore, Darmstadt, Germany) following incubation with specific antibodies.

### 4.4. Detection of mRNA Expression

To explore changes in mRNA expression levels, we used Blend Taq DNA polymerase (Toyobo, Osaka, Japan) and SYBR Fast qPCR Mix (Takara Bio Inc., Shiga, Japan), for PCR and qPCR, respectively. The following primers were used to amplify the target genes for RT-PCR and qPCR, respectively: c-FLIP (forward) 5′-CGGACTATAGAGTGCTGATGG-3′ and (reverse) 5′-GATTATCAGGCAGATTCCTAG-3′ and actin (forward) 5-GGCATCGTCACCAACTGGGAC-3′ and (reverse) 5′-CGATTTCCCGCTCGGCCGTGG-3′ for RT-PCR; c-FLIP (forward) 5′-CGCTCAACAAGAACCAGTG-3′ and (reverse) 5′-AGGGAAGTGAAGGTGTCTC-3′, STAMBPL1 (forward) 5′-GGGACCATCGCAGTGACAAT-3′ and (reverse) 5′-CCGACAGATGGAGCTTTGCT-3′, and actin (forward) 5′-CTACAATGAGCTGCGTGTG-3′ and (reverse) 5′-TGGGGTGTTGAAGGTCTC-3′ for qPCR. Actin was used as a reference gene to calculate the threshold cycle number (Ct) of each gene. Thereafter, ΔΔCt values were estimated.

### 4.5. Transfection

To overexpress or knockdown a given gene, a corresponding plasmid or siRNA was transfected into the cells using Lipofectamine2000 or Lipofectamine^®^ RNAiMAX Reagent (Invitrogen, Carlsbad, CA, USA), respectively.

### 4.6. Ubiquitination and Immunoprecipitation Assay

The ubiquitination and immunoprecipitation assay were performed as previously described [38]. Briefly, cells were sonicated on ice in RIPA lysis buffer containing 10 mM N-ethylmaleimide (Sigma-Aldrich) and 1 mM phenylmethylsulfonyl fluoride (Sigma-Aldrich). After incubation with the primary antibody overnight, PLUS-Agarose (Santa Cruz Biotechnology) was added to the lysate for 2 h at 4 °C. Thereafter, horseradish peroxidase-conjugated anti-Ub antibody (Enzo Life Sciences) was used for the ubiquitination assay under denaturing conditions. Protein–protein interactions were then explored via western blotting.

### 4.7. Xenograft Model

Male SCID mice purchased from JA Bio Inc. (Suwon, Republic of Korea) were inoculated subcutaneously at the flank with HCT116 (5 × 10^6^) cells. Thereafter, the mice were randomly distributed into four groups (*n* = 6 per group). This was followed by treatment with either vehicle (2% dimethyl sulfoxide [DMSO]/PBS), 10 mg/kg TBMS-1, or 3 mg/kg GST-TRAIL three times a week via intraperitoneal injection. At the end of the treatment period, tumor size was calculated using the formula [(length × width^2^)/2]. To verify apoptosis, we used an ApopTag Fluorescein In Situ Apoptosis Detection Kit (Merck Millipore) and detected terminal deoxynucleotidyl transferase (TdT) dUTP Nick-End Labeling (TUNEL)-positive staining. All the animal experiments were approved by the Keimyung University Ethics Committee (Approval number: KM-2022-03R1).

### 4.8. Analysis of Survival Rates, Correlation, and Expression in Patients with Renal Clear Carcinoma (RCC)

The overall survival data of patients with RCC were obtained using GEPIA2 (http://gepia2.cancer-pku.cn, 2018) based on The Cancer Gene Atlas (TCGA) cohort with high versus low expression [39]. Thereafter, STAMBPL1 and c-FLIP mRNA expression levels were obtained using UCSC Xena (https://xena.ucsc.edu/, 2019) by analyzing data from the TCGA cohort in the tumor tissues of patients with RCC [40].

### 4.9. Statistical Analysis

All statistical analyses were performed using SPSS software version 22.0 (SPSS Inc., Chicago, IL, USA). One-way ANOVA and post hoc comparisons (Student–Newman–Keuls test) were performed for group comparisons.

## 5. Conclusions

This study demonstrated that TBMS-1 sensitizes cancer cells to the apoptosis effect of TRAIL via STAMBPL1-dependent downregulation of c-FLIP. STAMBPL1 bound to c-FLIP and deubiqutylated c-FLIP protein. Therefore, TBMS-1 may be used as a sensitizer to TRAIL-induced apoptosis in cancer cells.

## Figures and Tables

**Figure 1 ijms-24-11840-f001:**
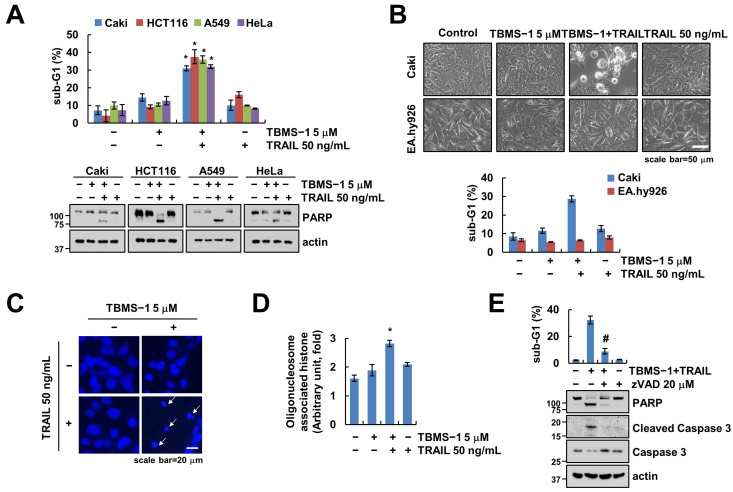
Effect of TBMS-1 on TRAIL-mediated apoptosis in human cancer cell lines. (**A**–**D**) Cancer cells (Caki, HCT116, A549, and HeLa; (**A**–**D**) and normal cells (EA.hy926; (**B**) were treated with 5 μM TBMS-1, 50 ng/mL TRAIL, or combinations for 24 h. Cell morphology was assessed using a microscope; scale bar: 50 μm (**B**). Condensation (**C**) and fragmentation (**D**) of nuclei were explored via DAPI staining and using a DNA fragmentation assay kit, respectively; scale bar: 20 μm. (**E**) Caki cells were pretreated with 20 μM z-VAD for 30 min, followed by treatment with 5 μM TBMS-1 and 50 ng/mL TRAIL for 24 h. The sub-G1 population and protein expression were measured via flow cytometry (**A**,**B**,**E**) and western blotting (**A**,**E**). The values shown represent the mean ± SD based on three independent experiments; * *p* < 0.01 relative to the control; # *p* < 0.01 relative to the TBMS-1 plus TRAIL treatment.

**Figure 2 ijms-24-11840-f002:**
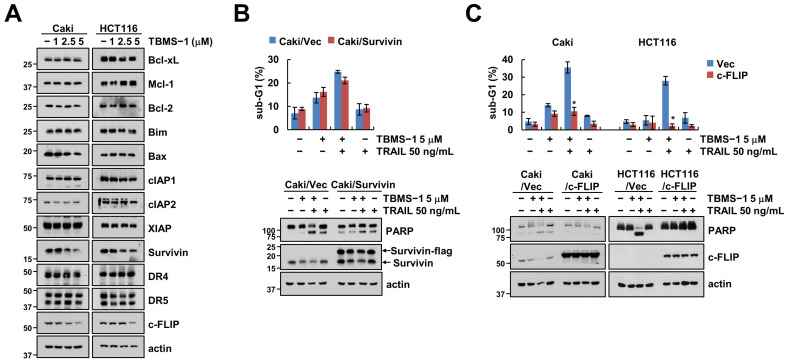
Downregulation of c-FLIP contributes to TBMS-1-induced TRAIL sensitization. (**A**) Cancer cells were treated with 1-5 μM TBMS-1 for 24 h. (**B**,**C**) Cancer cells were transfected with vector, survivin-flag (**B**) or c-FLIP (**C**) plasmid, followed by treatment with 5 μM TBMS-1, 50 ng/mL TRAIL, and both for 24 h. The sub-G1 population and protein expression levels were measured via flow cytometry (**B**,**C**) and western blotting (**A**–**C**). The values shown in graphs represent the mean ± SD based on three independent experiments; * *p* < 0.01 relative to TBMS-1 plus TRAIL treatment in vector-transfected cells.

**Figure 3 ijms-24-11840-f003:**
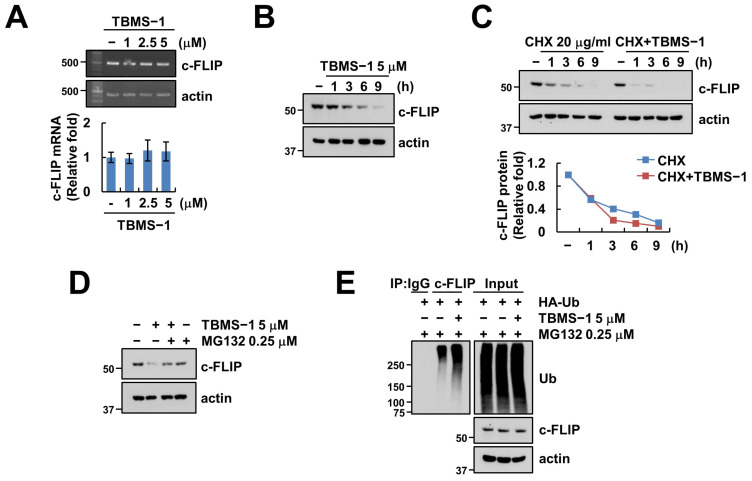
TBMS-1 ubiquitylates and degrades c-FLIP. (**A**) Examination of c-FLIP mRNA expression in Caki cells treated with 1–5 μM TBMS-1 for 24 h. (**B**) Examination of c-FLIP protein expression in Caki cells treated with 5 μM TBMS-1 for the indicated times. (**C**,**D**) Caki cells were pretreated with 20 μg/mL cycloheximide (CHX, **C**) or 0.25 μM MG132 (**D**) for 30 min, followed by treatment with 5 μM TBMS-1 for the indicated times. (**E**) Caki cells were transfected with HA-Ub plasmid and pretreated with 0.25 μM MG132, followed by treatment with 5 μM TBMS-1. The ubiquitination assay was performed using anti-c-FLIP antibody. mRNA and protein expression levels were determined via RT-PCR, qPCR (**A**), and western blotting (**B**–**E**), respectively.

**Figure 4 ijms-24-11840-f004:**
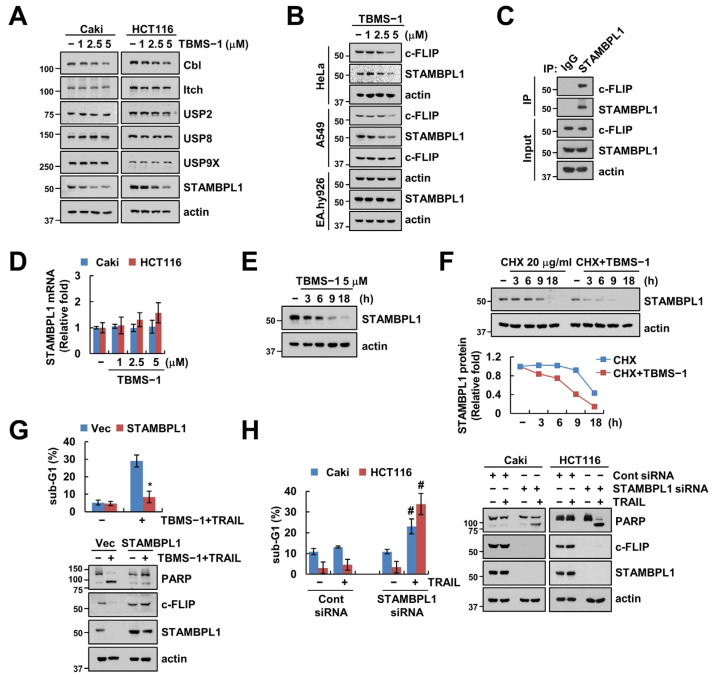
Involvement of STAMBPL1 in TBMS-1-mediated c-FLIP degradation and TRAIL sensitization. (**A**,**B**) Cancer and normal cells were treated with 1–5 μM TBMS-1 for 24 h. (**C**) Lysates from Caki cells were immunoprecipitated with anti-STAMBPL1 antibodies. (**D**) STAMBPL1 mRNA expression in cancer cells treated with 1–5 μM TBMS-1 for 24 h. (**E**) STAMBPL1 protein expression in Caki cells treated with 5 μM TBMS-1 for the indicated times. (**F**) Caki cells were pretreated with 20 μg/mL CHX for 30 min, followed by treatment with 5 μM TBMS-1 for the indicated times. (**G**) Caki cells were transfected with vector or STAMBPL1 plasmid, followed by treatment with a combination of 5 μM TBMS-1 and 50 ng/mL TRAIL for 24 h. (**H**) Caki cells were transfected with control siRNA or STAMBPL1 siRNA, followed by treatment with 50 ng/mL TRAIL for 24 h. Protein and mRNA expression levels were determined via western blotting (**A**–**C**,**E**–**H**) and qPCR (**D**), respectively. The sub-G1 population was measured via flow cytometry (**G**,**H**). The values in graphs represent the mean ± SD for three independent experiments; * *p* < 0.05 relative to TBMS-1 plus TRAIL treatment in vector-transfected cells.; # *p* < 0.01 relative to TRAIL treatment in control siRNA-transfected cells.

**Figure 5 ijms-24-11840-f005:**
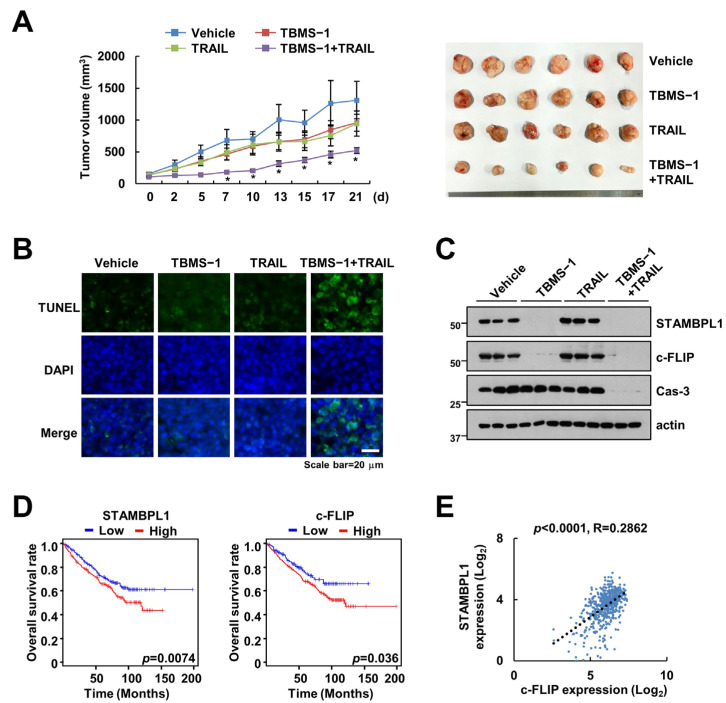
Suppression of tumor growth by TBMS-1 and TRAIL combination therapy in a xenograft model. (**A**–**C**) Mice bearing HCT116 cells were treated with 10 mg/kg TBMS-1, 3 mg/kg GST-TRAIL, or their combinations for 21 d. Growth curves and photographs of the tumors in mice (**A**). TUNEL (**B**) and western blot (**C**) assay results showing apoptosis and protein expression, respectively. (**D**) Overall survival rates of patients with RCC showing high STAMBPL1 and c-FLIP expression levels based on the TCGA database. (**E**) Correlation between the expression levels of STAMBPL1 and c-FLIP mRNA based on the TCGA patient cohort in Xena. * *p* < 0.01 compared to vehicle.

**Figure 6 ijms-24-11840-f006:**
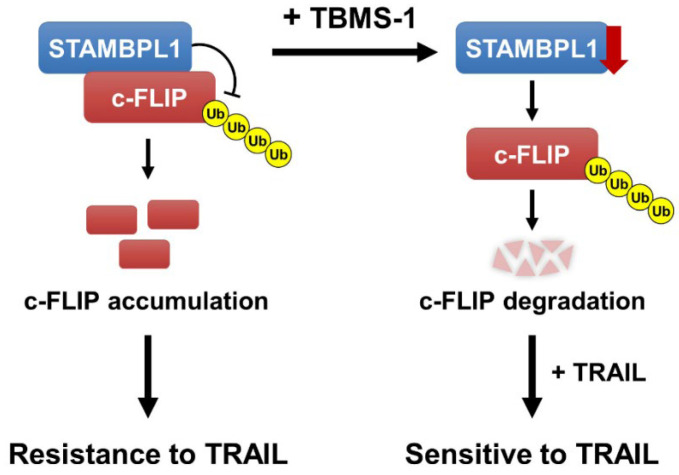
Schematic representation of the mechanisms used to overcome TRAIL resistance via TBMS-1-mediated c-FLIP degradation and STAMBPL1-dependent regulation of c-FLIP stabilization.

## Data Availability

The data presented in this study are available on request from the corresponding author.

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
