# Peer review of "Tubeimoside-1 Enhances TRAIL-Induced Apoptotic Cell Death through STAMBPL1-Mediated c-FLIP Downregulation"

_ijms, 2023, doi:10.3390/ijms241411840_

Round 1

Reviewer 1 Report

Dear Authors, 

Your manuscript is most of the time well design and executed, but a number of comments need to be addressed, before acceptance, to strengthen your manuscript. See below :

#1  Fig 1 : A TBMS1 dose effect in the 4 cell lines would be needed, combined or not to TRAIL given that the concentration used for A549 and HeLa may not be appropriate.

#2 It is also not clear why you still treat your cells using a Combo, which is TRAIL + TBMS1 at the same time, given that TBMS1 induces the loss of cFLIP in a time dependent manner. You could at least try a sequential treatment which are known to provide better treatment efficacy (see for ie Micheau et al British J Pharmacology 2003). You could for instance try to treat your cells for 6h with TBMS1 then add TRAIL and analyze the % of apoptosis (using Annexin V) 16 to 24 hours latter ! I would bet the results would even be better than the combo approach that you are using.

#3 : Fig 2 : Legend is missing. 

   Related to the point above, we cannot understand, given that this information is not described in the material and methods, how long have the cells been treated for ? 

Panel A  : When was the endpoint performed, 1h, 6h, 24h or 72 h after treatment ?????

Panel B : TBMS1 + TRAIL was using simultaneously (Combo) right ? 

            Description of the results for this figure is really poor.ie: panel A, you don’t even comment on the downregulation of cFLIP and Survivin, instead you commented on the results of the ectopic expression of these proteins… Rephrase your paragraph to provide a clear argumentation for the readers. At least present the data as they should be presented.

#4  Fig 3 : A control is missing here. The time-dependent degradation cFLIP with TBMS1 alone (as opposed to CHX and CHX + TBMS1, as shown panel B). Please include this information

#5 : Again the presentation and logical argumentation of the results felts short here. You should introduce why you have performed the STAMPL1 IP ???

#6 : Fig 4 : Panel A : shows the deregulation by 24 h of both c-FLIP and STABMPL1, which is also occurring in HeLa and A549, although in the latter no synergy with TRAIL was demonstrated (see figure 1).  

-        Like in figure #3 (see comment #4) , a control is missing panel D, you should show the  protein levels of STABMPL1 in the presence of TBMS1 alone.

-        There is also a discrepancy panel E regarding the %sub-G1 in STABMPL1 overexpressing cells vs PARP cleavage. How can these cells display sub-G1 while no PARP is cleaved ? Authors should use here another method to quantify apoptosis, ie Annexin V staining.

-        On the same vein panel F, I can easily understand that the deubiquitinylase STABMPL1 maybe important for cFLIP stability but it seems unlikely that its deficiency (shown here with a siRNA) would lead to the full degradation of cFLIP as shown in this panel, without allowing full killing of the corresponding cells after TRAIL- treatment. No more that 30-40% of the cells are killed here, one would except to get a better killing ! could you comment ?

-        Along the line, given that cFLIP is so well inhibited by the STABMPL1 siRNA, a control using TNFalpha, without chx, nor ciAP-inhibitors, should be used to demonstrate that indeed the effect solely mediated by cFLIP-induced degradation. In this case the % of apoptosis (Annexin V staining and PARP cleavage, + cFLIP) should be shown in cells expressing or not STAMBPL1 (same as fig 4 panel F).

#7  Throughout the manuscript and in particular in the result section presenting figure 4, the rational not sufficiently explained; In the latter case STABMPL1 is not presented. The corresponding sentence doesn’t allow the reader understand whether STABMPL1 is a Dub or an E3 ligase !

#8 Importantly and mandatory, given that you link STABMPL1 to the control of the degradation of cFLIP, you ought to demonstrate that the gain or loss of function in STABMPL1 ectopically expressing or silenced cells, is indeed related to changes in cFLIP ubiquitination, this is not shown. Please show.

#9 : One important aspect too of the work in missing. Yu have not shown the effect of you compound with respect to TRAIL receptor expression. That is something that you have done in the past for Honokiol and that is missing here, precluding proper discussion of your results. You must show by WB and flow cytometry, the expression levels of both TRAIL-R1 (DR4) and TRAIL-R2(DR5) as a kinetic after treatment of your cells (4 cell lines) with TBMS1. This is likely to help you discuss the different behavior of your cell lines to the combination and in addition may also help you consider in a different manner sequential treatments vs  COMBO therapies

#10  The discussion should be rewritten to  better discuss the similarities and the differences between henokiol and TBSM1 with regards to the synergy with TRAIL, and in particular why the approach is is not affected by ectopic expression of the surviving, whereas in the case of Henokiol it is. You should for instance discuss or comment on the differences of these  molecules? And propose a working hypothesis to explain this difference.

See comments to the authors. The writing is rather poor, and telegraphic ! Rational and argumentation should be better written to help the reader understand the strength and weaknesses of the manuscript.

Author Response

Dear,
We sincerely appreciate the time and effort of you and the referees spent in considering and evaluating our manuscript (ijms-2449036) entitled with “Tubeimoside-1 Enhances TRAILInduced Apoptotic Cell Death Through STAMBPL1-Mediated c-FLIP Downregulation” by So Rae Song et al., for publication in International Journal of Molecular Sciences. 

Having received the kind comments by reviewer, we revised our manuscript with attention to each of the comments by reviewer. We appreciate the reviewer very much, who raised the very important critiques to strengthen the claim of our manuscript. We have given very careful consideration to the suggestions and have revised our manuscript. We performed additional experiments and new information are incorporated in the revised version of our manuscript. 
We have responded all the comments by the referee point-by-point as follows in attached file.

Could you find attated file?

Sincerely yours

Reviewer 2 Report

"This paper has been prepared very reliably. I would like to suggest increasing the size of certain figures, particularly photographs Fig 1B, 1C, and 5B. Additionally, the legend for Figure 2 is missing. To enhance Figure 6, it would be beneficial to present a broader spectrum of relationships discussed in the paper. The authors may consider utilizing tools such as https://string-db.org/ to further support their findings."

Author Response

(The authors gave the same response as above.)

Reviewer 3 Report

The Research article topic “Tubeimoside-1 Enhances TRAIL-Induced Apoptotic Cell Death Through STAMBPL1-Mediated c-FLIP Downregulation” has very briefly summarized the information related its topic. The article needs major revisions by taking consideration of few points that incorporate needful changes, as the content provided is very precise enough.

11. In the introduction part, the author discusses the synergistic effect of Tubeimoside-1 (TBMS-1) and TRAIL in the sensitization of tumour cells following mechanistic pathway involving c-FLIP, a anti apoptotic regulator and STAMBPL1, a deubiquitinase. In the first paragraph, the author has very concisely described about the anti-cancer activity of TBMS-1. He/she should also provide detailed information about the “ Cytotoxic effects of TBMS-1 on different tumour cells” such as in cervical cancer, lung cancer and colorectal cancer and the effect of TBMS-1 in ameliorating mitochondrial membrane potential, upregulating cytochrome C and Bax/Bcl2 as well as involvement of ROS-AMPK signalling.

22. The author should also describe about the “TRAIL signalling pathway and its involvement in cancer therapy”.

33. The introductory part is quite short and should also incorporate the information about “Role of c-FLIP and STAMBPL-1 in tumour growth inhibition and its molecular pathway”.       

44.   High resolution of the figures should be provided as the picture quality of figures (such as in figure 1 B & 5 B)  are very poor.

55. The different concentrations are used for different drugs such as in TBMS-1, TRAIL and CHX. Authors need to mention why he chose such values and from where these values come from.

66. Lack of information in result section 2.3 & figure 3E, how TBMS-1 enhances c-FLIP ubiquitination.

77. The article should also provide detailed information about the proteins discussed in result section such as proteasome inhibitor, MG132, Cbl (Casitas B-lineage lymphoma) and DUBs (deubiquitinase).

88.  The article should also highlight about the “toxicity and pharmacokinetic analysis” of TBMS-1 as it is prone to degradation by GI tract after oral administration.

99.  The concluding part is very concisely discussed and needs to be elaborated.

Minor editing of English language required

Author Response

(The authors gave the same response as above.)

Reviewer 4 Report

This manuscript investigated the mechanism of combining tubeimoside-1 (TBMS-1), a traditional Chinese medicinal herb, with TRAIL to induce apoptotic cell death in various cancer cell lines. Fig 1 provides data supporting this combination using 4 different cancer cell lines. Interestingly, treatment of a non-cancer cell line EA.hy926 did not result in increased sub-G1. Fig. 2 provides data showing a decrease in survivin and c-FLIP with TBMS-1 but only overexpression of c-FLIP blocked apoptosis. Reduction of c-FLIP by TBMS-1 was due to increased protein degradation (Fig. 3). Further investigation identified the STABMPL1 (reduced by TBMS-1) as a DUB that can regulate c-FLIP levels (Fig. 4). Experiments in xenograft mice further support use of this combination  (Fig. 5).

Overall, the data is very supportive for using TBMS-1 (reduces STABMPL1 resulting in reduction of c-FLIP) in combination with TRAIL. Experiments are well done and clearly explained. There are a few questions and suggestions with the intention of improving the quality.

1.    Would be interesting to know in EA.hy926 cells if TBMS-1 reduces STABMPL1 and c-FLIP, as in cancer cells.

2.    Some figure legends are misplaced or missing. Fig. 2 legend needs to be completed as instructions are included. Explanation of higher MW survivin in Caki/survivin in panel B? Misplaced in Fig. 3 legend. Therefore, appears that Fig. 3 legend is missing.

3.    Fig. 3D when first mention STABMPL1 should be placed in Fig. 4.

4.    Materials and methods: None of the primer sequences provided for mRNA expression matched to appropriate genes, as determined by BLAST search. Need to add correct primer sequences used in experiments or need to repeat using correct primers.

5.    Materials and methods: number of HCT cells injected in SCID mice?

Minor English editing corrections required. 

Author Response

(The authors gave the same response as above.)

Round 2

Reviewer 1 Report

Dear Authors, let me first of all thank you for your revised version.

As you will read below, I still have a few concerns that I hope you will be prepared to address.

In particular I am very impressed by loss of cFLIP expression after STAMBPL1 siRNA. Yet this loss of expression of cFLIP doesn’t appear to restore TNF-induced cell death, as asked in comment N°10, although it ought to be the case. This maybe due to the absence of TNFR1 in the cell line that you have used. Given that you didn’t comment on the cell line tested, I guessed it could have been the Caki cells (See also comment on the cell line below). Could you please check this in HCT116 cells +/- siRNA STAMBPL1 (without cycloheximide) reported to be sensitive to TNFalpha (https://doi.org/10.1158/1541-7786.MCR-16-0329) ? You should also include a positive control HCT116 TNF and CHX. You may also check Fas ligand sensitivity (without CHX) as these 3 membrane bound receptors rely on caspase-8 activation , inhibited by cFLIP, to trigger cell death.

As commented above I came to wonder which cells you were working with  :  Caki-1 Οr CaSki - CRL-1550cells ? Please add this information into the material and methods section (include also HCT116, A549 etc…).

It would be nice to discuss the limitations of your study, in particular if you are not willing to address my last comments (si STAMPBPL1 in HCT116 to see if TNF is inducing Killing). If not could you explain why TNFalpha is not killing these cells ? Along the line, you findings may also indicate that something else is at work to explain the gain of apoptosis function, as the sequential treatment you have tested (Caki cells), is even less efficient than the combined treatment ! Please discuss in this section of the manuscript. Would it be the same for A549 or HCT116 ?

The conclusion is also too short.

English should be corrected by a native English speaker.

see above

Author Response

Dear,
We sincerely appreciate the time and effort of you and the referees spent in considering and evaluating our manuscript.

Having received the kind comments by reviewer, we revised our manuscript with attention to each of the comments by reviewer. We appreciate the reviewer very much, who raised the very important critiques to strengthen the claim of our manuscript. We have given very careful consideration to the suggestions and have revised our manuscript. We performed additional experiments and new information are incorporated in the revised version of our manuscript. Also, we modified English language through editing services. We have responded all the 
comments by the referee point-by-point as follows in attached file.

Could you find attated file?

Sincerely yours

Reviewer 3 Report

Comments are addressed.

Manuscript could be accepted for publication.

Moderate editing of English language required

Author Response

Thank you for your comments.

As your comments, we modified English language through editing services. 

Round 3

Reviewer 1 Report

Thanks for addressing my comments

none

Reviewer 3 Report

Comments are addressed. MS is suitable for publication.

English check is still required.